# Effects of Exogenous Methyl Jasmonate on Metabolism and Soil Activity in *Chrysanthemum morifolium*

**DOI:** 10.3390/plants14193026

**Published:** 2025-09-30

**Authors:** Guimei Tang, Fan Zhao, Xiaoling Xiao, Yingshu Peng, Yuxia Zhou, Li Zhang, Jilong Yang, Yuanzhi Xiao, Yang Liu, Weidong Li, Guolin Huang

**Affiliations:** 1Institute of Horticulture, Hunan Academy of Agricultural Sciences, Changsha 410125, China; tangguimei@hunaas.cn (G.T.); zhaofan2023@hunaas.cn (F.Z.); xiaoxiaoling@hunaas.cn (X.X.); yspeng@hunaas.cn (Y.P.); yuxiaz75@163.com (Y.Z.); zhli911@hunaas.cn (L.Z.); yjloog@hunaas.cn (J.Y.); xiaoyuanzhi458@hunaas.cn (Y.X.); 2Yuelu Mountain Laboratory, Changsha 410125, China; 3Hunan Provincial Key Laboratory for Germplasm Innovation and Utilization of Landscape Plants and Flowers, Changsha 410125, China

**Keywords:** methyl jasmonate, continuous cropping obstacles, *Chrysanthemum morifolium*, soil microbiota, untargeted metabolomics

## Abstract

Challenges significantly hinder the sustainable cultivation of tea chrysanthemum, leading to imbalances in soil nutrients, the accumulation of allelopathic phenolic acids, reduced enzymatic activity, and disruptions in rhizosphere microbial communities. To explore potential mitigation strategies, this study systematically evaluated the integrative effects of exogenous methyl jasmonate (MeJA, 0–400 (μmol L^−1^)) on both soil environmental parameters and plant growth performance under continuous cropping conditions. The results revealed that treatment with 100 (μmol L^−1^) MeJA significantly enhanced plant height, canopy width, flower number, and fresh flower weight. Concurrently, it improved soil organic matter content, the available nitrogen levels, and redox stability while increasing the activity of key enzymes, including polyphenol oxidase, urease, and catalase. Notably, this treatment markedly reduced the accumulation of allelopathic phenolic acids, such as p-hydroxybenzoic acid and vanillic acid. High-throughput sequencing further demonstrated that 100 (μmol L^−1^) MeJA optimized the composition of soil microbial communities, increasing the abundance of beneficial taxa, such as nitrogen-fixing and phosphate-solubilizing bacteria, while suppressing pathogenic fungi. Metabolomic analysis showed that this concentration of MeJA activated stress-resistance metabolic pathways involving flavonoids and terpenoids while downregulating degradation-related processes, thereby supporting enhanced plant resilience at the metabolic level. Collectively, these findings demonstrate that an appropriate concentration of exogenous MeJA can effectively alleviate continuous cropping obstacles in *Chrysanthemum morifolium*, providing both theoretical insights and practical guidance for its eco-friendly and efficient cultivation.

## 1. Introduction

Chrysanthemum (*Chrysanthemum morifolium*), one of the most popular ornamental plants worldwide, holds a prominent position in the global cut flower market, ranking among the top four species in commercial value [1]. Owing to its rich varietal diversity, vibrant coloration, and unique floral morphology, chrysanthemum is widely favored by consumers in both cut flower and potted plant markets. In recent years, the rapid increase in consumer demand for high-quality floral products has driven the industrial-scale expansion of chrysanthemum cultivation. However, this accelerated development has also intensified reliance on monoculture practices, giving rise to a series of challenges that threaten the sustainability of the chrysanthemum industry [2,3]. In addition to MeJA-based regulation, other biotechnological approaches—such as tissue culture, transgenic breeding, application of plant growth regulators (e.g., salicylic acid, gibberellins), and inoculation with beneficial microbes—have also been explored to improve chrysanthemum growth and stress tolerance under intensive cultivation [4]. Among these, continuous cropping obstacles have emerged as a major bottleneck. Prolonged monoculture leads to the progressive degradation of soil ecological functions, characterized by nutrient depletion, reduced enzymatic activity, proliferation of soilborne pathogens, and accumulation of allelopathic compounds, such as phenolic acids [5]. These cumulative effects impair plant growth, reduce ornamental quality, and may result in severe physiological disorders, including dwarfism, leaf rot, and floral malformations, ultimately stripping plants of their commercial value [6]. Moreover, the soil imbalance associated with continuous cropping renders chrysanthemums more vulnerable to pathogenic infections, such as Fusarium wilt disease (FWD), a devastating condition that drastically compromises both yield and quality, posing serious economic risks to growers [2]. Addressing continuous cropping obstacles has therefore become a critical and urgent priority in chrysanthemum cultivation and management.

Continuous cropping obstacles, also referred to as soil fatigue, describe the phenomenon where repeated cultivation of the same crop or closely related species leads to suppressed plant growth. The primary causes of this phenomenon include nutrient imbalances, disruption of microbial communities, and the accumulation of phytotoxic compounds [7,8,9]. This phenomenon not only inhibits normal plant growth but also significantly damages the ecological functions of the soil. One major mechanism underlying continuous cropping obstacles is allelopathy, in which chemicals secreted by plants, particularly phenolic acids, accumulate in the rhizosphere, negatively affecting both root systems and surrounding microbial communities [10,11]. Additionally, prolonged monoculture results in significant changes to the soil microbial community, typically characterized by a reduction in beneficial microbes and an increase in pathogenic microbes, which makes crops more susceptible to diseases [12]. Research has shown that as the duration of continuous cropping increases, chrysanthemum yield gradually declines, while disease incidence increases year by year. After 15 years of continuous cropping, the yield of chrysanthemum was only 19.80% of the initial yield, and disease incidence reached 100%. Furthermore, the number of bacteria and actinomycetes peaked after 3 years of planting and then gradually decreased, while soil fungalization progressively worsened. The microbial composition of continuous cropping soils underwent significant changes, with a decrease in beneficial microbial groups and an increase in pathogenic microbial populations, further exacerbating the degradation of soil health [13].

Studies on the rhizosphere fungal communities of chrysanthemums under continuous cropping conditions revealed that while diversity and abundance were not significantly affected, the composition of the fungal communities changed notably. Specifically, there was a reduction in beneficial fungi (e.g., *Beauveria*) and an increase in pathogenic fungi (e.g., *Fusarium*, *Verticillium*, *Rhizoctonia*), which further explained the decline in chrysanthemum health and productivity under continuous cropping conditions [14]. In studies on the continuous cropping of African daisy, it was found that with an increase in the number of cropping years, the bulk density and pH of the soil increased, while the levels of available phosphorus and potassium decreased. However, the content of organic matter and available nitrogen increased. This indicates that continuous cropping not only alters the microbial community but also leads to the deterioration of soil physicochemical properties [15]. Moreover, continuous cropping is closely linked to the deterioration of soil physicochemical properties. Imbalances or deficiencies in essential nutrients, such as carbon, nitrogen, and phosphorus, not only hinder plant nutrient uptake but also weaken the soil’s intrinsic ability to self-restore, further affecting plant health and growth [16].

MeJA is a vital phytohormone ubiquitously present in plants, playing a central role in regulating growth, stress responses, and defense mechanisms [17]. Among various biotic stresses faced by plants, pathogenic micro-organisms are among the most prevalent and destructive. To cope with such challenges, plants have evolved complex, multi-layered defense systems, including both physical barriers and chemical defenses [18]. A prominent example is the hypersensitive response (HSR) through which plants reinforce their cell walls and synthesize secondary metabolites to restrict pathogen invasion [19]. Exogenous application of MeJA or its functional analogs—such as jasmonoyl-β-D-glucoside, jasmonoyl-gentiobiose, and hydroxylated jasmonoyl-β-D-glucoside—can strongly induce systemic acquired resistance (SAR), thereby enhancing plant resilience to pathogens and other biotic stressors [20,21,22]. Although various MeJA derivatives, such as jasmonoyl-β-D-glucoside, jasmonoyl-gentiobiose, and hydroxy-jasmonoyl-β-D-glucoside, have been identified, the evidence of their direct roles in alleviating continuous cropping obstacles remains scarce.

Increasing evidence suggests that MeJA is also effective in alleviating continuous cropping obstacles not only by modulating rhizosphere microbial communities but also by influencing the accumulation and transformation of allelochemicals, such as phenolic acids [23,24,25]. Previous studies in rice have demonstrated that exogenous MeJA treatment activates the phenylpropanoid pathway, significantly enhancing the expression and activity of key enzymes, such as phenylalanine ammonia-lyase (PAL) and cinnamate 4-hydroxylase (C_4_H). This leads to increased accumulation of phenolic acids, including p-coumaric acid, ferulic acid, vanillic acid, and caffeic acid [26]. These compounds play dual roles in rhizosphere ecology: they function as defense-related metabolites and signaling molecules that suppress pathogens or regulate microbial interactions; yet, their excessive accumulation under continuous monoculture conditions may aggravate allelopathic stress and microbial imbalance.

Moreover, MeJA is reported to influence soil enzyme activities either directly through altered root exudation or indirectly via the activation of microbe-mediated metabolic pathways. Enzymes such as PAL, peroxidase (POD), and polyphenol oxidase (PPO), which are frequently induced by MeJA, are not only involved in plant defense but also in the degradation or detoxification of allelochemicals in soil. Thus, MeJA may contribute to the mitigation of continuous cropping obstacles by regulating phenolic acid metabolism, enhancing enzyme-mediated soil functions, and promoting a beneficial shift in microbial community structure. Based on these insights, it can be hypothesized that MeJA alleviates continuous cropping stress in *chrysanthemum* through a synergistic mechanism involving (1) modulation of phenolic acid biosynthesis and turnover, (2) stimulation of soil enzymatic activities related to organic matter cycling and allelochemical detoxification, and (3) rebalancing of rhizosphere microbial communities.

For instance, root drenching with MeJA has been shown to significantly boost strawberry yield, increase soil organic matter content, and enhance the abundance of specific beneficial microbial taxa [27]. Additionally, MeJA application has been found to mitigate postharvest decay in peaches caused by *Rhizopus stolonifer* by enhancing the activity of defense-related enzymes, such as chitinase, β-1,3-glucanase, phenylalanine ammonia-lyase, polyphenol oxidase, and peroxidase [28]. Moreover, MeJA confers multifaceted defense mechanisms against abiotic stresses like salinity. These include regulating the balance of inorganic ions and organic osmolytes, elevating the levels of endogenous hormones, and enhancing antioxidant enzyme activities, all of which contribute to improved stress tolerance [21].

However, the physiological and biochemical effects of exogenously applied MeJA vary considerably depending on its application concentration. Numerous studies have demonstrated that different MeJA concentrations can elicit distinct outcomes in terms of stress resistance regulation, growth promotion, and secondary metabolite accumulation. For instance, Ahmadi et al. [29] reported that foliar application of 0.1 mM MeJA significantly increased relative water content, soluble sugar levels, and photosynthetic rate in rapeseed seedlings, thereby mitigating the detrimental effects of NaCl-induced salt stress. MeJA has also been used to alleviate self-incompatibility in *Camellia oleifera*. In one study, application of 1000 μmol·L^−1^ MeJA markedly enhanced pollen germination and pollen tube elongation, ultimately leading to a higher fruit set rate [30]. The synthesis of volatile compounds and the activity of aroma-related enzymes in tobacco have been shown to vary significantly with MeJA concentrations ranging from 0 to 100 μM, indicating a strong dosage-dependent effect [31]. Low concentrations of MeJA (10–50 μM) have also been reported to markedly enhance phytochemical accumulation and antioxidant capacity in *Dracocephalum polychaetum* [32]. Callus cultures of an apple hybrid treated with 1, 10, 100, and 1000 μM MeJA exhibited increased growth at lower concentrations but significant growth inhibition at the highest concentration [33]. Furthermore, treatment of *Mentha piperita* with 1 mM MeJA, in combination with varying zinc levels, led to increased levels of phenolic compounds and elevated activities of antioxidant enzymes [34].

Collectively, these findings underscore the critical role of MeJA concentration in modulating plant physiological responses and growth regulation. Given the diversity of plant species and their exposure to varying stress conditions, optimizing the concentration of MeJA application is essential to maximizing its beneficial effects. Such optimization not only enhances its practical efficacy but also provides a solid theoretical foundation for its targeted application in crop management.

Although previous studies have highlighted the multifaceted physiological roles of MeJA in mitigating plant stress responses, its specific mechanisms in alleviating continuous cropping obstacles in chrysanthemum, as well as the optimal concentration for application, remain insufficiently explored. As a major ornamental and economically valuable crop, *chrysanthemum* cultivation is increasingly constrained by continuous-cropping-related issues, which represent a critical bottleneck to its sustained and efficient production. Thus, there is an urgent need to develop effective mitigation strategies. In this study, we applied a series of MeJA treatments at varying concentrations to systematically evaluate its regulatory effects on agronomic traits, plant growth parameters, soil enzyme activities, and rhizosphere microbial community structure in continuously cropped chrysanthemum. The concentration range of 0–400 (μmol L^−1^) was selected based on both the existing literature and our preliminary experiments. Previous studies have demonstrated that low concentrations of MeJA (10–50 μM) can enhance plant growth and secondary metabolite accumulation, while concentrations exceeding 400 (μmol L^−1^) tend to inhibit growth and lead to oxidative stress [29,30,31,32,33,34]. This concentration range provides a clear gradient of physiological responses, ensuring a robust comparison of MeJA’s dose-dependent effects on chrysanthemum under continuous cropping stress. The objective was to elucidate the physiological and ecological mechanisms by which MeJA alleviates continuous cropping stress and to determine the most effective application concentration. The findings of this research are expected to provide both theoretical insights and practical guidance for the prevention and management of continuous cropping obstacles, thereby contributing to the sustainable development of the chrysanthemum industry.

## 2. Materials and Methods

### 2.1. Experimental Materials

*Chrysanthemum morifolium* is rich in flavonoids and other bioactive compounds. It shows strong antioxidant and medicinal properties. Because of its stable composition and wide availability, it is often chosen as an ideal material for functional studies and experimental analyses. Plants were cultivated at the *Chrysanthemum* Base of the Hunan Institute of Horticultural Research.

In this study, plants were grown under full sunlight, with at least 6 h of direct exposure per day. Temperature was maintained at 18–25 °C. Soil with good drainage and high organic content was used. The pH was adjusted to 6.5. Balanced fertilizer was applied regularly. Moisture was controlled by drip irrigation to keep soil wet but not waterlogged. Relative humidity was kept between 60% and 80%. These conditions provided optimal growth and reduced environmental interference with experimental results.

Six concentrations of MeJA were applied: 0 (μmol L^−1^) (control, CK), 10 (μmol L^−1^) (T1), 50 (μmol L^−1^) (T2), 100 (μmol L^−1^) (T3), 200 (μmol L^−1^) (T4), and 400 (μmol L^−1^) (T5). Each treatment included three biological replicates, giving a total of 18 groups. Before transplantation, seedlings were immersed in MeJA solution for 10 min at the assigned concentration. After transplantation, foliar spraying was performed every 5 days. Each plant received 50 mL solution per treatment. All treatments ended 45 days after transplantation.

### 2.2. Plant Sampling

Sampling was conducted at the full-bloom stage of *Chrysanthemum morifolium*. To ensure representativeness, five healthy plants were selected from each treatment group as biological replicates. Entire plant structures were included, covering roots, stems, leaves, and flowers. Samples from each organ were evenly collected to avoid bias from a single tissue type.

### 2.3. Soil Sampling

At the flowering stage of *Chrysanthemum morifolium*, rhizosphere soil (1–2 mm layer) was collected using the gentle brushing method. The plant was uprooted; non-rhizosphere soil was shaken off; and a sterile brush was used to collect soil from the root surface, minimizing root disturbance. The sampling depth was 0–15 cm, corresponding to the root zone. Six biological replicates were collected per treatment, with three samples mixed for physicochemical analysis and three kept separate for microbial diversity and enzyme activity analyses. For physicochemical analysis, samples were air-dried, ground, and sieved through a 100 mesh. For microbial analysis, samples were frozen in liquid nitrogen and stored at −80 °C for subsequent sequencing, enzyme activity determination, and microbial community analysis. This standardized sampling method, consistent depth, defined replication, and clear mixing strategy ensure the reproducibility and scientific reliability of the data collection process.

### 2.4. Agronomic Trait Measurement

At the full-bloom stage of *Chrysanthemum morifolium*, agronomic traits were recorded under different MeJA treatments (0, 10 (μmol·L^−1^), 50 (μmol·L^−1^), 100 (μmol·L^−1^), 200 (μmol·L^−1^), and 400 (μmol·L^−1^)). For each treatment, five representative plants were randomly selected for measurement. The recorded traits included plant height (vertical distance from the soil surface to the highest point of the plant); canopy diameter (average of east–west and north–south diameters, measured using the cross method) [35].

Number of inflorescences per plant (all fully opened inflorescences); inflorescence diameter (mean of the maximum diameters of three randomly selected inflorescences per plant); fresh flower weight per plant (total fresh weight of all open inflorescences); and the number of ray and tubular florets (from one randomly chosen inflorescence per plant) [36]. The collected data were subsequently subjected to statistical analysis.

### 2.5. Determination of Soil Physicochemical Properties and Phenolic Acids

Soil redox potential was measured using a fully automatic FJA-6 depolarization redox potential analyzer (Nanjing Soil Instrument Co., Nanjing, China) following the manufacturer’s instructions [37]. Approximately 20 g of fresh soil was placed in a beaker, and the electrode was inserted at a depth of 5 cm. The values were recorded after stabilization for 3–5 min. Soil pH was determined with a multi-parameter meter (pH/ORP/conductivity/dissolved oxygen, SX751, SANXIN, Shanghai, China) following the agricultural industry standard NY/T 1377-2007 [38]. Soil and distilled water were mixed at a 1:2.5 (*w*/*v*) ratio, shaken for 30 min, and allowed to equilibrate before measurement.

Alkaline nitrogen was determined using the Kjeldahl distillation method. Briefly, 5 g of air-dried soil was digested with alkaline KMnO_4_ solution, followed by distillation and titration with standard H_2_SO_4_ solution to calculate the nitrogen content. The available phosphorus was measured using the NaOH fusion–UV spectrophotometric method. Soil samples were fused with NaOH at 750 °C, extracted with HCl, and the absorbance was determined at 700 nm.

The available potassium was determined via NaOH fusion followed by flame photometry [39]. Approximately 5 g of soil was fused with NaOH, dissolved in distilled water, filtered, and analyzed using a flame photometer (Model 6400A, Shanghai Precision Instruments, Shanghai, China).

Soil enzyme activities were determined as follows. Urease activity was assayed using the indophenol blue colorimetric method [40]. In brief, 5 g of fresh soil was incubated with 10% urea solution at 37 °C for 24 h. The released NH_4_^+^ was quantified colorimetrically at 578 nm. Catalase activity was measured via UV spectrophotometry [41]. Fresh soil (5 g) was incubated with 0.3% H_2_O_2_ solution, and the decomposition of H_2_O_2_ was determined by monitoring absorbance at 240 nm. Polyphenol oxidase (PPO) activity was determined spectrophotometrically [42]. Soil samples were incubated with catechol solution, and absorbance was measured at 430 nm after 30 min to calculate enzyme activity. Urease activity was also verified via the phenol–sodium hypochlorite colorimetric method [43], where ammonium ions were extracted and quantified at 578 nm to cross-validate the results.

Phenolic acids in rhizosphere soil—including p-hydroxybenzoic acid, vanillic acid, p-coumaric acid, cinnamic acid, ferulic acid, and salicylic acid—were quantified using high-performance liquid chromatography (HPLC, 1260 Infinity II, Agilent Technologies, Santa Clara, CA, USA) [39]. Soil extracts were prepared with methanol:water (80:20, *v*/*v*), filtered through a 0.22 μm membrane, and separated on a C18 reversed-phase column (250 × 4.6 mm, 5 μm). The mobile phase consisted of methanol and 0.1% formic acid (gradient elution), with a flow rate of 1.0 (mL·min^−1^) at 30 °C. Detection was performed at 280 nm.

### 2.6. LC–MS Untargeted Metabolomics Analysis

Samples were collected at the full-bloom stage of *Chrysanthemum morifolium*. Entire plants (roots, stems, leaves, and flowers) were used. Five plants per treatment were selected as replicates. Untargeted metabolomics was performed using an LC–MS approach. An Ultimate 3000 high-performance liquid chromatography system was coupled with an Orbitrap Exploris 480 mass spectrometer (Thermo Fisher Scientific, Waltham, MA, USA). A Waters Acquity UPLC HSS T3 column (1.8 μm, 2.1 × 100 mm; Waters Corporation, Milford, MA, USA) was employed. The mobile phases were 0.1% formic acid in water and 0.1% formic acid in acetonitrile, under both positive and negative ion modes. The injection volume was 1 μL.

Mass spectrometry was operated in DDA mode, with a scan range of m/z 67–1000. ESI ion source settings included a spray voltage of ±2500–3500 V, gas flow rates of 50/10/1 arb, ion transfer tube temperature of 325 °C, and vaporizer temperature of 350 °C. Raw *.raw* files were processed using Compound Discoverer software (CD, version 3.3, Thermo Fisher Scientific, Waltham, MA, USA). The processing included peak extraction, alignment, and correction of m/z and retention time. Alignment was performed with a 5 ppm mass tolerance and a 0.2 min retention time window. The peaks were extracted with a signal-to-noise ratio ≥ 3, intensity ≥ 10^5^, and intensity deviation ≤ 30%.

Metabolites were annotated against mzCloud (Thermo Fisher Scientific), MZvault (Thermo Fisher Scientific), and ChemSpider (Royal Society of Chemistry) databases. After peak area normalization, principal component analysis (PCA) and Spearman correlation were used to evaluate reproducibility. Metabolite classification and pathway annotation were performed using KEGG [44], HMDB [45], and LIPID MAPS [46]. Differential metabolites were identified via OPLS-DA modeling. The screening criteria were fold change >1, *p* < 0.05, and VIP > 1. Significant pathway enrichment was assessed using the hypergeometric test.

### 2.7. High-Throughput Sequencing for Soil Microbial Diversity

Soil samples were sent to Biomarker Technologies Co., Ltd. (Beijing, China) for high-throughput amplicon sequencing of bacterial 16S rRNA and fungal ITS regions. Total genomic DNA was extracted from soil samples treated with different MeJA concentrations (0, 10 (μmol L^−1^), 50 (μmol L^−1^), 100 (μmol L^−1^), 200 (μmol L^−1^), and 400 (μmol L^−1^)) using the TGuide S96 Magnetic Soil/Fecal DNA Kit (Tiangen Biotech, Beijing, China), following the manufacturer’s instructions.

The bacterial 16S rRNA gene V3–V4 region was amplified with primers 338F (5′-ACTCCTACGGGAGGCAGCA-3′) and 806R (5′-GGACTACHVGGGTWTCTAAT-3′). The fungal ITS1 region was amplified with primers ITS1-F (5′-CTTGGTCATTTAGAGGAAGTAA-3′) and ITS2 (5′-GCTGCGTTCTTCATCGATGC-3′). PCR products were verified via agarose gel electrophoresis, purified with the Omega DNA Purification Kit (Omega Bio-tek, Norcross, GA, USA), and sequenced on the Illumina NovaSeq 6000 platform (Illumina Inc., San Diego, CA, USA) with a 2 × 250 bp paired-end strategy.

Raw sequences were processed using USEARCH (v10.0; Robert C. Edgar, Tiburon, CA, USA) for quality control and clustered into OTUs at 97% similarity. Taxonomic annotation of OTUs/ASVs was performed with the QIIME2 naive Bayes classifier (version 2023.9; QIIME2 Development Team, https://qiime2.org/), based on the SILVA database (v138.1), with a confidence threshold of 70%. Alpha diversity was assessed using QIIME2. Beta diversity was visualized via principal coordinate analysis (PCoA).

Differences in bacterial community abundance and diversity were analyzed using one-way ANOVA. Differential taxa were identified using linear discriminant analysis (LDA) with effect size (LEfSe). All sequencing data analyses were carried out on the BMKCloud online platform (https://www.biocloud.net, accessed on 20 April 2025).

### 2.8. Data Analysis

All experimental data were processed and analyzed using Excel 2019 and SPSS 26.0. One-way analysis of variance (ANOVA) was applied to compare differences among treatments in soil physicochemical properties, enzyme activities, phenolic acid contents, microbial structures, and plant metabolic changes. When ANOVA indicated significant differences, Duncan’s multiple range test (DMRT) was used as the post hoc test for multiple comparisons. Different letters above the bars or within the tables indicate significant differences among treatments. The significance threshold was set at *p* < 0.05.

## 3. Results

### 3.1. Effects of Different Exogenous MeJA Concentrations on the Growth Characteristics of Chrysanthemum morifolium

To determine the appropriate concentration of exogenous MeJA for alleviating replanting obstacles in *Chrysanthemum morifolium*, growth traits were systematically investigated at the maturity stage. Plant height, canopy diameter, number of inflorescences per plant, inflorescence diameter, fresh flower weight per plant, and the numbers of ray and tubular florets were recorded under different treatments (Figure 1).

Significant differences were observed among treatments. At 100 (μmol L^−1^), plant height, canopy diameter, and fresh flower weight were all higher than those in the control (0, CK). Ray floret number increased significantly, while tubular floret number decreased. These results indicated that 100 (μmol L^−1^) MeJA was most favorable for plant growth and the improvement of inflorescence quality. However, further increases in MeJA concentration (200 (μmol L^−1^) and 400 (μmol L^−1^)) did not enhance these positive effects. In some traits, a declining trend was observed. In particular, at 400 (μmol L^−1^) (T5), both growth and inflorescence quality were suppressed, suggesting that excessive MeJA may trigger adverse responses in plants.

### 3.2. Effects of Different MeJA Concentrations on Soil Physicochemical Properties and Enzyme Activities

Soil physicochemical properties and enzyme activities are key factors influencing plant growth and microbial activity. They directly determine the stability of the rhizosphere ecosystem and the efficiency of nutrient supply. The results of this study showed that exogenous MeJA had a significant regulatory effect on soil properties and functional enzyme activities. The most pronounced responses were observed at 100 (μmol L^−1^), where nutrient availability, redox status, and metabolic activity reached optimal levels, reflecting improved ecological stability.

In terms of physicochemical properties, 100 (μmol L^−1^) MeJA treatment significantly increased soil organic matter and alkaline nitrogen contents (*p* < 0.05). Specifically, organic matter content in 100 (μmol L^−1^) MeJA treatment reached ~7.5% (Figure 2A), significantly higher than CK and other treatments. Alkaline nitrogen (Figure 2B) increased to nearly 200 mg/kg in T3, indicating enhanced mineralization and rapid nitrogen release, which improved nitrogen supply for plant uptake. For the available potassium (Figure 2C) and available phosphorus (Figure 2D), no significant differences were found among treatments, but the 100 (μmol L^−1^) MeJA treatment maintained stable levels without the decline or stress effects observed under higher concentrations. Moreover, soil pH was significantly elevated in the 100 (μmol L^−1^) MeJA treatment (Figure 2E), reaching a slightly alkaline value of 7.51, higher than other groups. Most importantly, the redox potential in T3 (Figure 2F) was maintained at a moderate level, between the strong oxidative state of 10 (μmol L^−1^) MeJA treatment and the strong reductive state of 400 (μmol L^−1^) MeJA treatment, forming a “neutral to mildly reductive” condition.

In terms of soil functional enzyme activities, 100 (μmol L^−1^) MeJA treatment also showed significant positive effects. Polyphenol oxidase (PPO) activity in the 100 (μmol L^−1^) MeJA treatment was significantly lower than that in CK and low-concentration treatments (Figure 2G). In contrast, catalase (CAT) activity reached the highest level in T3 (Figure 2H). For carbon- and nitrogen-related enzymes, sucrase activity was significantly enhanced in 100 (μmol L^−1^) MeJA treatment (Figure 2I), and urease activity was also markedly increased (Figure 2J). Collectively, these results indicated that 100 (μmol L^−1^) MeJA treatment effectively improved soil physicochemical properties and strengthened soil metabolic activity, demonstrating a coordinated regulatory effect.

### 3.3. Effects of Different MeJA Concentrations on Soil Phenolic Compounds

Continuous cropping obstacles are mainly caused by the accumulation of harmful substances in soil, especially phenolic acids. These compounds inhibit root development through allelopathy, thereby restricting plant growth [10]. To assess the effect of MeJA, five major phenolic acids were quantified in soils treated with different concentrations. The results showed that MeJA significantly influenced their accumulation, and distinct trends were observed with increasing concentrations.

With the 100 (μmol L^−1^) MeJA treatment, MeJA markedly suppressed the accumulation of p-hydroxybenzoic acid, vanillic acid, and salicylic acid (Figure 2K,L,O). However, with the 400 (μmol L^−1^) MeJA treatment, their levels increased again. For p-coumaric acid, no clear difference was found between low concentrations (T1–T3) and the control (CK), although a slight increase was observed. With the 200 (μmol L^−1^) MeJA treatment, p-coumaric acid content increased sharply to the highest level (Figure 2M). When the concentration was further raised to 400 (μmol L^−1^) MeJA treatment, its level declined significantly. In contrast, ferulic acid content was consistently higher in all MeJA treatments compared with CK, but changes among the concentrations were relatively small (Figure 2N).

In summary, exogenous MeJA with the 100 (μmol L^−1^) MeJA treatment effectively suppressed the accumulation of most phenolic acids in soil. At higher concentrations, this inhibitory effect did not strengthen, and in some cases, it even reversed, leading to increased accumulation.

### 3.4. High-Throughput Sequencing Analysis of Soil Microbial Diversity

Soil microbial communities are essential for maintaining soil health, promoting plant growth, and improving fertility. In this study, high-throughput sequencing was used to evaluate the effects of different concentrations of MeJA on soil microbial diversity. Both fungal and bacterial communities were analyzed at taxonomic levels (phylum and family), and their ecological functions were interpreted. The results demonstrated that MeJA significantly altered the composition and relative abundance of soil micro-organisms. With the 100 (μmol L^−1^) MeJA treatment, the microbial community structure was optimized, with enrichment of beneficial taxa and suppression of potential pathogens.

#### 3.4.1. Effects on Fungal Community Structure

The bar plots of relative abundance showed that the fungal community was dominated by *Ascomycota* and *Basidiomycota*. *Ascomycota* accounted for more than 60% in all treatments and remained stable across concentrations, highlighting its ecological stability and role in organic matter degradation, nutrient cycling, and rhizosphere regulation (Figure 3). *Basidiomycota* decreased under 100 (μmol L^−1^) MeJA treatment but increased at higher concentrations (200 (μmol L^−1^) MeJA treatment, 400 (μmol L^−1^) MeJA treatment), suggesting stimulation under high MeJA concentration. *Mortierellomycota* remained stable, while *Glomeromycota* was enriched at T3, indicating promotion of arbuscular mycorrhizal fungi (AMF), which enhance nutrient uptake and rhizosphere competitiveness. Rare taxa, including *Chytridiomycota*, *Olpidiomycota*, and unclassified fungi, showed dynamic shifts, implying that MeJA may influence rare fungal groups through rhizosphere interactions.

At the family/genus level (Figure 4), clear treatment effects were observed. *Penicillium* (Figure 4A) and *Humicola nigrescens* (Figure 4G) were enriched with the 100 (μmol L^−1^) MeJA treatment, both known for their strong capacity to degrade cellulose, lignin, and complex organics, enhancing soil fertility and producing antibiotics against pathogens. Conversely, *Ustilaginales* (Figure 4D), a group containing plant pathogens, declined significantly under 100 (μmol L^−1^) MeJA treatment but remained higher at other concentrations. This suggests pathogen suppression under the 100 (μmol L^−1^) MeJA treatment. *Didymellaceae* (Figure 4E) and *Sordariomycetes* (Figure 4C) were maintained at moderate levels in T3, which may help preserve balance by preventing excessive dominance of harmful fungi. *Eurotiomycetes* (Figure 4F) remained abundant in 100 (μmol L^−1^) MeJA treatment but decreased under high MeJA, indicating the sensitivity of some beneficial fungi to excessive treatment. In contrast, *Kernia* (Figure 4B) and *Kernia geniculotricha* (Figure 4H) decreased with the 100 (μmol L^−1^) MeJA treatment, suggesting reduced ecological competitiveness, indirectly favoring beneficial groups.

#### 3.4.2. Effects on Bacterial Community Structure

At the phylum level (Figure 5), bacterial responses were concentration-dependent. T3 significantly increased the abundance of *Methylomirabilota* and *Gemmatimonadota*, while most other phyla remained relatively stable across treatments.

At the genus/family level (Figure 6), multiple beneficial taxa were enriched with the 100 (μmol L^−1^) MeJA treatment. *Tumebacillus* (Figure 6A) increased significantly, enhancing organic matter decomposition and nutrient cycling. *Dehalococcoidia* (Figure 6B) was also enriched, known for degrading harmful compounds, such as organohalides, thereby improving soil health. *Lysinibacillus* (Figure 6C) increased, contributing to disease resistance and nitrogen utilization. *Kribbella* (Figure 6D) and *Saprospiraceae* (Figure 6E) were also enhanced, promoting organic matter degradation and nutrient release. *Erysipelotrichales* (Figure 6F) rose in abundance, supporting soil health and root development. *Planococcaceae* (Figure 6G) increased, reinforcing organic matter turnover and disease suppression. Finally, *Kribbella karoonenesis* (Figure 6H) was enriched, further optimizing nutrient cycling and plant growth conditions.

Overall, the 100 (μmol L^−1^) MeJA treatment optimized the soil microbial ecosystem by enriching beneficial fungi and bacteria while suppressing pathogenic taxa. This concentration enhanced organic matter degradation, nutrient cycling, and root symbiosis, thereby creating a healthier rhizosphere environment. These findings provide theoretical support for the use of exogenous MeJA in regulating soil microbial communities and improving the growth environment of medicinal chrysanthemums.

### 3.5. LC–MS Untargeted Metabolomics Analysis of Chrysanthemum morifolium Under Different Exogenous Methyl Jasmonate Treatments

Untargeted metabolomic profiling of *Chrysanthemum morifolium* treated with different concentrations of MeJA was performed using liquid chromatography–mass spectrometry (LC–MS). The results showed that MeJA significantly altered multiple metabolic pathways. Volcano plot analysis revealed distinct differences in metabolite changes across treatments. In T1 (10 (μmol L^−1^)), 82 metabolites were upregulated and 136 downregulated; in T2 (50 (μmol L^−1^)), 11 were upregulated and 201 downregulated; in T3 (100 (μmol L^−1^)), 244 were upregulated and 173 downregulated; in T4 (200 (μmol L^−1^)), 140 were upregulated and 108 downregulated; and in T5 (400 (μmol L^−1^)), 215 were upregulated and 190 downregulated (Figure 7).

The KEGG classification of differential metabolites (Figure 8) indicated a clear bidirectional regulatory pattern in the plant’s response to MeJA. Key defensive pathways were activated, while certain degradative or stress-related pathways were selectively suppressed. This reallocation of metabolic resources supported both optimization of secondary metabolite production and maintenance of physiological homeostasis.

At low concentration (10 (μmol L^−1^)), pathways related to flavonoid degradation and fluorobenzoate degradation were downregulated (Figure 8F). This indirectly promoted accumulation of secondary metabolites, particularly flavonoids. Although the number of upregulated pathways was small, enhanced activity in aromatic metabolism linked to antioxidant defense was already evident (Figure 8A). Thus, even low concentrations of MeJA initiated a modest defensive response. At 50 (μmol L^−1^), the scope of metabolic reprogramming broadened. Along with continued suppression of degradation pathways, several amino acid metabolism routes (e.g., tyrosine metabolism), antibiotic biosynthesis, and secondary metabolite biosynthetic pathways were downregulated (Figure 8G). This suggested a metabolic reallocation in which non-essential processes were reduced to prioritize defense-related metabolism. Concurrently, upregulated pathways increased, including antimicrobial compound biosynthesis and phenylpropanoid biosynthesis (Figure 8B), reflecting enhanced defense capacity. At 100 (μmol L^−1^), MeJA induced the strongest metabolic response. Upregulated pathways were significantly enriched in key modules, such as flavonoid, alkaloid, phenylpropanoid, and terpenoid biosynthesis, together with energy metabolism, amino acid biosynthesis, and carbon–nitrogen metabolism (Figure 8C). Meanwhile, pathways such as flavonoid degradation, cyanoamino acid metabolism, and indole alkaloid biosynthesis were suppressed (Figure 8H). These results suggested that while functional secondary metabolism was strongly activated, overactivation of stress-related or degradative pathways was avoided, ensuring a balanced response to external stimuli.

In summary, MeJA regulation of metabolic pathways showed a concentration-dependent response. At low concentrations (10–50 (μmol L^−1^)), plants maintained metabolic homeostasis by suppressing degradative pathways and slightly activating defense metabolism. At the medium concentration (100 (μmol L^−1^)), multiple core pathways were activated and stress-related ones suppressed, achieving the optimal coordination of defense and growth. At high concentrations (200–400 (μmol L^−1^)), metabolism was inhibited, with widespread downregulation that may impair growth and development (Figure 8D,E,I,J). Therefore, 100 (μmol L^−1^) can be regarded as the optimal concentration of MeJA for metabolic regulation, particularly under replanting obstacles and disease stress in chrysanthemum, providing a scientific basis for enhancing plant stress resistance.

## 4. Discussion

This study systematically analyzed the regulatory effects of MeJA on the agronomic traits of *Chrysanthemum morifolium* under continuous cropping conditions by setting different concentration gradients of exogenous MeJA treatment. The results clarified the optimal concentration of MeJA for alleviating *Chrysanthemum morifolium* continuous cropping obstacles. The findings indicated that treatment with appropriate concentrations of MeJA significantly enhanced the agronomic traits of *Chrysanthemum morifolium*, including plant height, crown width, number of flowers per plant, inflorescence diameter, fresh weight of flowers per plant, and the number of ligulate flowers, while effectively reducing the number of tubular flowers. Among the different concentrations tested, 100 (μmol L^−1^) MeJA showed the most significant promoting effect. However, as the MeJA concentration further increased to 200 (μmol L^−1^) and above, all the agronomic indicators showed varying degrees of decline. This suggests that excessively high concentrations of MeJA may have a suppressive effect on *Chrysanthemum morifolium*’s growth. Therefore, the choice of exogenous MeJA concentration exhibits a threshold effect. That is, the appropriate concentration (100 (μmol L^−1^)) significantly promotes *Chrysanthemum morifolium*’s growth and improves inflorescence quality, whereas higher concentrations lead to a decline in its regulatory effects.

### 4.1. Regulation of Soil Nutrients by MeJA

The physicochemical properties of soil, particularly its nutrient status, play a crucial role in plant growth [47]. This study found that exogenous MeJA treatment significantly increased the organic matter and alkali-hydrolyzable nitrogen content in the rhizosphere soil, with the most noticeable effect observed at 100 μmol/L concentration. This result supports the positive role of MeJA in optimizing the soil nutrient environment. MeJA may regulate the pH of rhizosphere soil and enhance its redox capacity, thereby promoting the decomposition of organic matter in the soil and increasing the release and cycling of nutrients. Additionally, MeJA application may further enhance soil nutrient transformation and nutrient absorption by plants by increasing microbial community activity. This mechanism indicates that MeJA, as a plant hormone, can optimize the rhizosphere environment by regulating soil chemical properties and biological activity, thus providing favorable nutrient conditions for plant growth in complex soil environments.

However, although MeJA can significantly improve soil nutrition in some cases, the increase in soil nutrients may also have potential negative effects on plant growth. For example, nitrogen-induced soil acidification can reduce microbial diversity and alter community composition [48]. Future research should further explore the effects of MeJA treatment on *Chrysanthemum morifolium* growth at different stages and across different soil types to identify the optimal timing for its application.

### 4.2. Degradation Effect of MeJA on Phenolic Acids

Phenolic acids are important components of continuous cropping obstacles, and their long-term accumulation in the soil can significantly inhibit plant growth, especially root development [49]. Studies show that MeJA can significantly promote the degradation of soil phenolic acids, with the effect being particularly pronounced at 100 (μmol L^−1^) concentration. This degradation effect of MeJA may be achieved by activating the soil enzyme system, particularly through increased activity of polyphenol oxidase and peroxidase. The accelerated degradation of phenolic acids creates a healthier rhizosphere environment for root growth, alleviating soil degradation caused by continuous cropping.

However, the degradation of phenolic acids is not solely dependent on the direct action of MeJA; it is likely closely related to changes in the soil microbial community structure [50]. MeJA significantly altered the microbial community in the rhizosphere, increasing the abundance of beneficial micro-organisms while inhibiting the growth of certain pathogens. Micro-organisms play a crucial role in organic matter decomposition and phenolic acid degradation [51]. Future research could explore in more detail the interaction mechanism between MeJA and microbial communities, further clarifying its effect on the degradation of phenolic acids.

### 4.3. The Effects of Different Concentrations of MeJA on Different Crops

In addition to the effects observed in *Chrysanthemum morifolium*, studies have shown that MeJA exerts similar effects on other crops, although with distinct differences in the concentration-dependent responses and microbial community interactions. For example, in strawberry, MeJA treatment has been shown to increase both fruit yield and resistance to pathogens, particularly at concentrations of 100 (μmol L^−1^) [27]. Similarly, in peach trees, MeJA application at 1 mM significantly improved pollen germination and fruit set rate [28]. However, in these crops, higher MeJA concentrations (such as (μmol L^−1^)) led to growth inhibition, indicating a concentration-dependent effect that is consistent with the findings in *chrysanthemums*.

In our study, the 100 (μmol L^−1^) concentration of MeJA exhibited the most significant positive effects on *Chrysanthemum morifolium* growth, particularly in terms of promoting agronomic traits and enhancing stress tolerance. However, higher concentrations (e.g., 200 (μmol L^−1^) and above) led to varying degrees of decline in all measured agronomic indicators, suggesting a threshold effect for MeJA in *chrysanthemums*, similar to that seen in other crops. The concentration-dependent effects of MeJA in chrysanthemum are thus in line with its role in other species, but the response in *chrysanthemum* appears to be more sensitive at higher concentrations.

Moreover, the response of the rhizosphere microbial community to MeJA in *chrysanthemums* differed from that observed in strawberries and peaches. In those crops, MeJA primarily promoted the growth of beneficial microbes, such as nitrogen-fixing bacteria and phosphate-solubilizing bacteria, while inhibiting certain pathogens. In *chrysanthemums*, however, MeJA treatment not only promoted beneficial micro-organisms but also had a more pronounced effect on microbial community restructuring, especially by inhibiting specific pathogens associated with continuous cropping. This suggests that MeJA’s impact on the microbial community may vary depending on the plant species, providing unique insights into how MeJA modulates microbial interactions in different crops under continuous cropping stress.

### 4.4. Microbe–Plant Synergistic Effect

Soil micro-organisms play a crucial role in maintaining soil health and promoting plant growth [52]. Research has shown that MeJA not only optimizes the plant growth environment by improving the physicochemical properties of soil but also promotes soil ecological health by regulating the microbial community [53]. In this study, exogenous MeJA significantly increased the abundance of beneficial micro-organisms in the rhizosphere soil, such as nitrogen-fixing bacteria and phosphate-solubilizing bacteria. This effect was most prominent at the 100 (μmol L^−1^) concentration, where the proliferation of beneficial microbes reached its highest level.

Conversely, MeJA treatment significantly inhibited the growth of certain potential pathogens, particularly fungi and bacteria associated with continuous cropping obstacles. These changes suggest that MeJA not only effectively optimizes the structure of the microbial community but may also enhance plant health through improved plant–microbe symbiosis.

The role of micro-organisms in nutrient transformation and plant disease control has garnered increasing attention. By enhancing the functional capacity of the microbial community, MeJA improves soil disease resistance and nutrient cycling, thereby creating a healthier growth environment for plants [54]. However, the impact of MeJA on microbial communities may vary depending on its concentration, indicating that future studies should further explore the regulatory mechanisms of MeJA at different concentrations. Additionally, research should investigate its effects across various continuous cropping soil types and tea chrysanthemum cultivars to optimize its application.

### 4.5. Mechanism of Plant Metabolic Regulation

This study demonstrates that the application of MeJA significantly promoted the growth and stress resistance of *Chrysanthemum morifolium*, especially under continuous cropping conditions. Our findings are consistent with previous studies, indicating that MeJA enhances plant quality and resistance by regulating both primary and secondary metabolic pathways.

Targeted metabolomics analysis revealed that MeJA treatment significantly promoted the synthesis of secondary metabolites, such as flavonoids, terpenoids, and phenolic acids, particularly at a concentration of 100 (μmol L^−1^). At this concentration, plant height, crown width, and fresh flower weight per plant significantly increased (Figure 1). This indicates that MeJA application can effectively stimulate the growth and flowering of *Chrysanthemum morifolium*, especially under the challenge of continuous cropping obstacles, significantly improving the plant’s growth rate and stress tolerance. MeJA enhances plant tolerance of environmental stress by regulating antioxidant metabolic pathways [55]. Specifically, MeJA likely boosts the activity of antioxidant enzymes, such as superoxide dismutase (SOD) and catalase (CAT), thereby improving the plant’s ability to cope with oxidative stress. Meanwhile, MeJA also promotes the biosynthesis of secondary metabolites like flavonoids, which possess both antioxidant and disease resistance properties, further strengthening the plant’s defensive capabilities.

In addition to its antioxidant effects, exogenous MeJA treatment at 100 (μmol L^−1^) also significantly promoted the accumulation of secondary metabolites in the plant, particularly flavonoids and terpenoids. These compounds play important roles in plant defense and stress responses. The accumulation of flavonoids, such as quercetin and kaempferol, helps mitigate oxidative stress and enhances plant resistance. Moreover, MeJA-induced synthesis of terpenoids, known for their antimicrobial and signaling properties, further helps plants resist pathogens under continuous cropping conditions.

In conclusion, MeJA alleviates continuous cropping stress in *Chrysanthemum morifolium* through a multifaceted mechanism (1) by activating antioxidant pathways via secondary metabolites, such as flavonoids; (2) regulating phenolic acid accumulation and its detoxification; and (3) modulating root exudates, improving the structure of the rhizosphere microbial community. These effects collectively promote plant growth, enhance stress resistance, and improve soil health, providing new theoretical insights for mitigating continuous cropping obstacles.

### 4.6. Experimental Limitations and Future Research Directions

In this study, we explored the regulatory effects of MeJA on *Chrysanthemum morifolium* under continuous cropping stress. However, there are several experimental limitations that may affect the generalizability and long-term effects of the results. First, the study primarily relied on pot experiments, which, while allowing for precise control of environmental variables, may not fully represent real-field conditions due to simplified soil environments and limited pot volume. Therefore, future studies should extend the research to field trials to validate the applicability of these results under natural conditions. Additionally, this study focused on the short-term effects, especially the immediate responses of plant growth and microbial communities. However, the long-term impacts of MeJA on plant growth, soil health, and microbial community dynamics remain unclear. Thus, extending the duration of the study to evaluate the long-term effects of MeJA on ecological systems, particularly on the stability of soil microbial communities and sustained plant growth, is a key direction for future research.

Future research should also explore the synergistic effects of MeJA and soil amendments, such as organic fertilizers and microbial inoculants, to optimize soil health and plant growth. Moreover, further studies could focus on the molecular mechanisms of MeJA, particularly its regulation of key gene expression related to stress resistance. Genomic and transcriptomic analyses could reveal how MeJA affects genes involved in plant defense, antioxidant pathways, and secondary metabolite biosynthesis, providing deeper insights into its molecular role in plant physiological regulation. Through these future research directions, we can gain a more comprehensive understanding of MeJA’s physiological mechanisms, optimize its agricultural applications, and advance its use in crop management and sustainable agriculture.

## 5. Conclusions

This study demonstrates the role of exogenous MeJA in alleviating replanting obstacles of *Chrysanthemum morifolium* and reveals a clear concentration-dependent response, with 100 (μmol L^−1^) identified as the optimal threshold for growth promotion and soil ecological stability. At this concentration, MeJA significantly improved soil physicochemical properties and enzyme activities, enriched beneficial microbial taxa while suppressing potential pathogens, and enhanced plant growth and metabolic activity, thereby achieving coordinated optimization across the soil–microbe–plant system. In contrast, higher concentrations exerted inhibitory effects, highlighting the need for careful dosage management. Future work should focus on elucidating the molecular mechanisms of MeJA signaling, testing its adaptability under diverse ecological conditions, and assessing long-term ecological impacts while also exploring its synergistic potential with other agronomic practices or biocontrol strategies. Together, these efforts will provide a more comprehensive framework for the sustainable cultivation of medicinal *chrysanthemum*.

## Figures and Tables

**Figure 1 plants-14-03026-f001:**
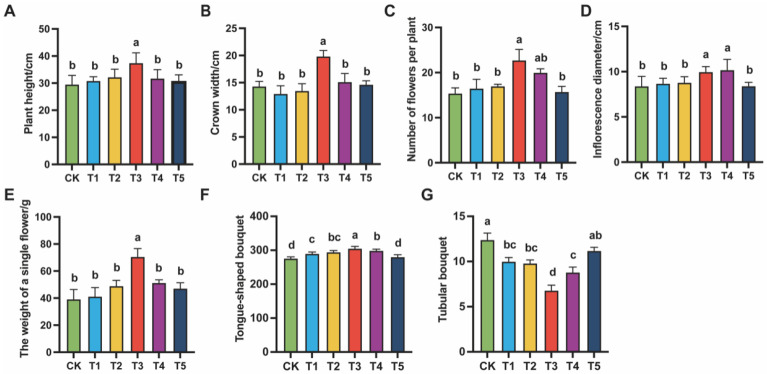
Effects of different concentrations of MeJA on the agronomic traits of *Chrysanthemum morifolium*. The treatment groups included the control (CK) and exogenous MeJA treatments at concentrations of 10 (μmol L^−1^) (T1), 50 (μmol L^−1^) (T2), 100 (μmol L^−1^) (T3), 200 (μmol L^−1^) (T4), and 400 (μmol L^−1^) (T5). (**A**) Plant height; (**B**) Crown width; (**C**) Number of flowers per plant; (**D**) Inflorescence diameter; (**E**) Fresh flower weight per plant; (**F**) Number of ray florets; (**G**) Number of tubular florets. Each data point represents the mean of five individual plants (N = 5). Different letters indicate statistically significant differences among treatment groups (*p* < 0.05).

**Figure 2 plants-14-03026-f002:**
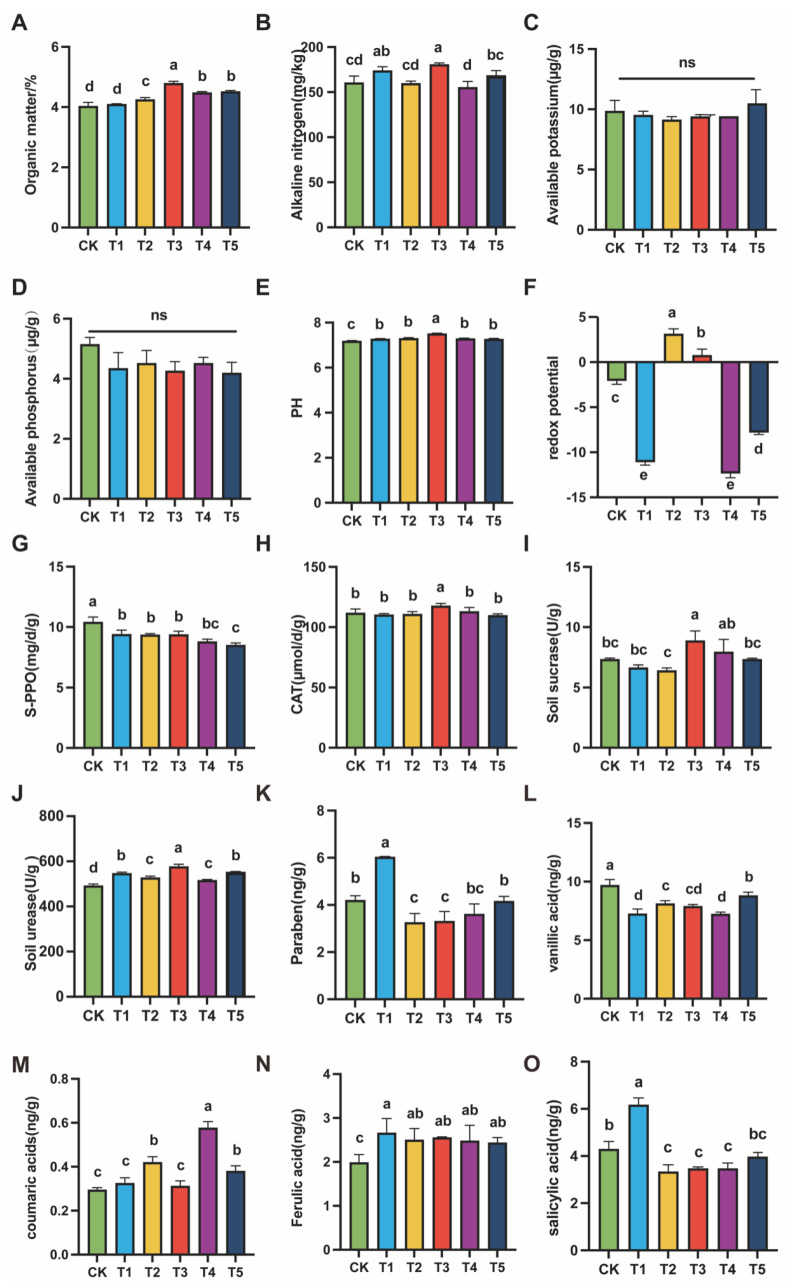
Soil physicochemical properties, enzyme activity indicators, and phenolic acid content under different concentrations of MeJA treatment. The treatment groups included the control (CK) and exogenous MeJA treatments at concentrations of 10 (μmol L^−1^) (T1), 50 (μmol L^−1^) (T2), 100 (μmol L^−1^) (T3), 200 (μmol L^−1^) (T4), and 400 (μmol L^−1^) (T5). (**A**) Organic matter content; (**B**) Alkali-hydrolyzable nitrogen; (**C**) Available phosphorus; (**D**) Available potassium; (**E**) pH value; (**F**) Redox potential; (**G**) Polyphenol oxidase activity; (**H**) Catalase activity; (**I**) Invertase activity; (**J**) Urease activity; (**K**) p-Hydroxybenzoic acid; (**L**) Vanillic acid; (**M**) p-Coumaric acid; (**N**) Ferulic acid; (**O**) Salicylic acid. Data are presented as the mean of three biological replicates. Different letters indicate statistically significant differences among treatment groups (*p* < 0.05).

**Figure 3 plants-14-03026-f003:**
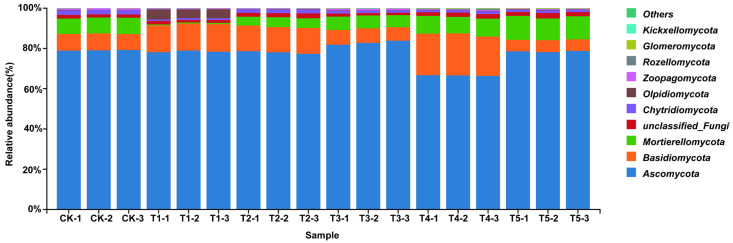
Bar chart of fungal phylum-level abundance in soil under different concentrations of exogenous MeJA treatment. The treatment groups include the control (CK) and exogenous MeJA treatments at concentrations of 10 (μmol L^−1^) (T1), 50 (μmol L^−1^) (T2), 100 (μmol L^−1^) (T3), 200 (μmol L^−1^) (T4), and 400 (μmol L^−1^) (T5).

**Figure 4 plants-14-03026-f004:**
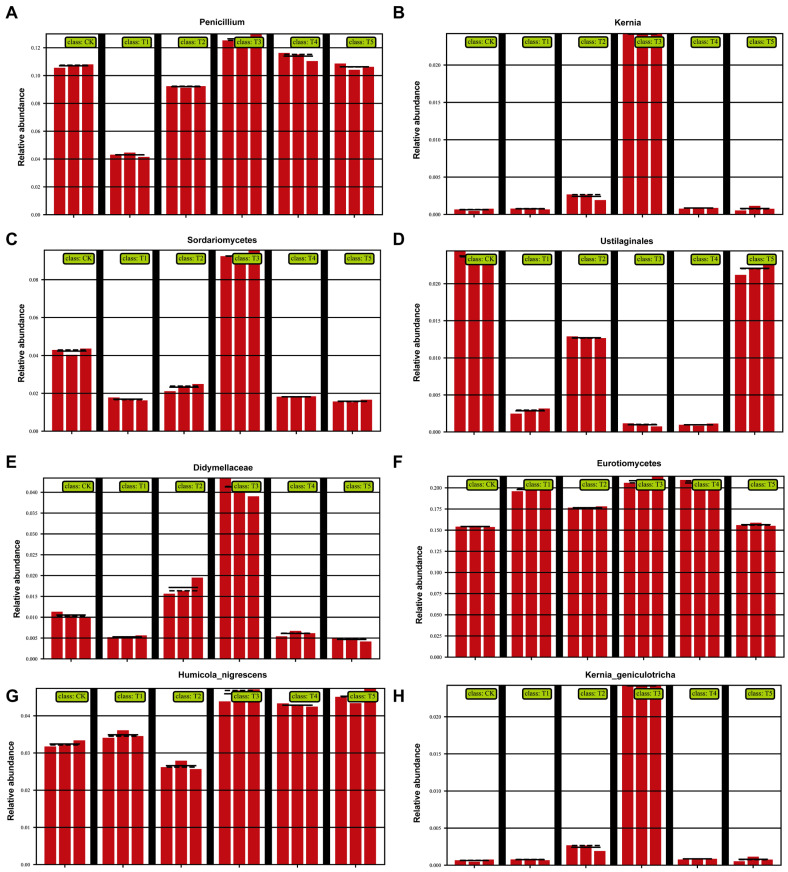
Effects of different concentrations of MeJA on the relative abundance of dominant soil fungal taxa at the family level (CK: 0 (μmol L^−1^), T1: 10 (μmol L^−1^), T2: 50 (μmol L^−1^), T3: 100 (μmol L^−1^), T4: 200 (μmol L^−1^), T5: 400 (μmol L^−1^)). (**A**) *Penicillium*; (**B**) *Kernia*; (**C**) *Sordariomycetes*; (**D**) *Ustilaginales*; (**E**) *Didymellaceae*; (**F**) *Eurotiomycetes*; (**G**) *Humicola nigrescens*; (**H**) *Kernia geniculotricha*.

**Figure 5 plants-14-03026-f005:**
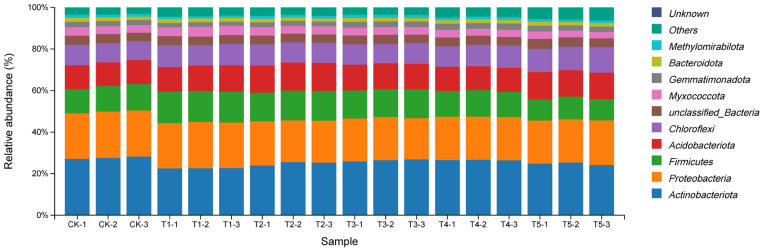
Bar chart of bacterial phylum-level abundance in soil under different concentrations of exogenous MeJA treatment. The treatment groups include the control (CK) and exogenous MeJA treatments at concentrations of 10 (μmol L^−1^) (T1), 50 (μmol L^−1^) (T2), 100 (μmol L^−1^) (T3), 200 (μmol L^−1^) (T4), and 400 (μmol L^−1^) (T5).

**Figure 6 plants-14-03026-f006:**
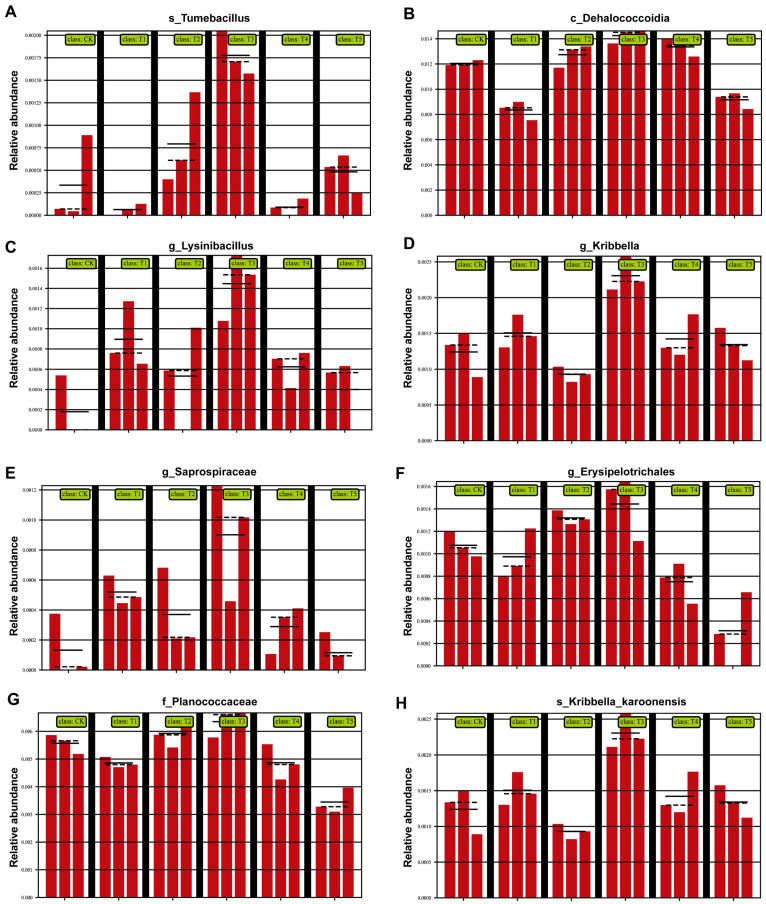
Effects of different concentrations of MeJA on the relative abundance of dominant soil bacterial taxa at the family level (CK: 0 (μmol L^−1^), T1: 10 (μmol L^−1^), T2: 50 (μmol L^−1^), T3: 100 (μmol L^−1^), T4: 200 (μmol L^−1^), T5: 400 (μmol L^−1^)). (**A**) *s_Tumebacillus*; (**B**) *c_Dehalococcoidia*; (**C**) *g_Lysinibacillus*; (**D**) *g_Kribbella*; (**E**) *g_Saprospiraceae*; (**F**) *g_Erysipelotrichale*; (**G**) *f_Planococcaceae*; (**H**) *s_Kribbella karoonenesis* (the black vertical lines serve as visual separators between treatment groups, while the highest point of the red bars represents the maximum value on the vertical axis, indicating the upper limit of the coordinate scale).

**Figure 7 plants-14-03026-f007:**
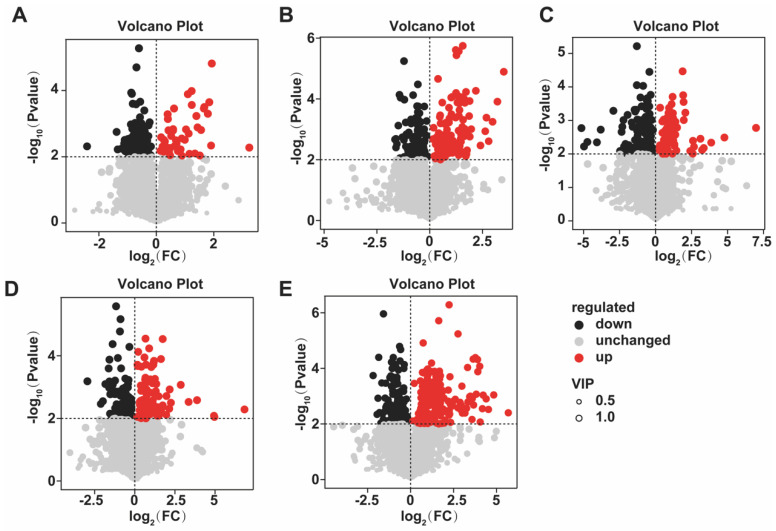
Volcano plots of differential metabolites under different concentrations of MeJA treatment. The treatment groups include the control (CK) and exogenous MeJA treatments at concentrations of 10 (μmol L^−1^) (T1), 50 (μmol L^−1^) (T2), 100 (μmol L^−1^) (T3), 200 (μmol L^−1^) (T4), and 400 (μmol L^−1^) (T5). (**A**) CK vs. T1; (**B**) CK vs. T2; (**C**) CK vs. T3; (**D**) CK vs. T4; (**E**) CK vs. T5.

**Figure 8 plants-14-03026-f008:**
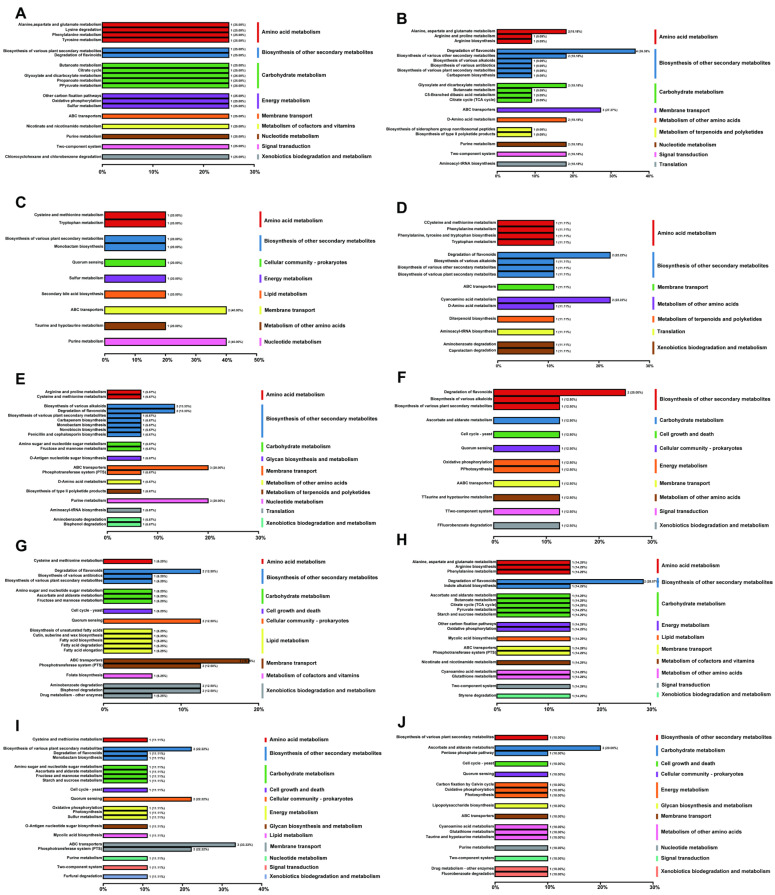
KEGG classification of upregulated and downregulated differential metabolites under different concentrations of MeJA treatment. The treatment groups include the control (CK) and exogenous MeJA treatments at concentrations of 10 (μmol L^−1^) (T1), 50 (μmol L^−1^) (T2), 100 (μmol L^−1^) (T3), 200 (μmol L^−1^) (T4), and 400 (μmol L^−1^) (T5). (**A**–**E**) KEGG classification of upregulated differential metabolites: (**A**) CK vs. T1; (**B**) CK vs. T2; (**C**) CK vs. T3; (**D**) CK vs. T4; (**E**) CK vs. T5. (**F**–**J**) KEGG classification of downregulated differential metabolites: (**F**) CK vs. T1; (**G**) CK vs. T2; (**H**) CK vs. T3; (**I**) CK vs. T4; (**J**) CK vs. T5.

## Data Availability

The metabolomics data generated and analyzed during this study were deposited in the MetaboLights database under the accession number MTBLS12478. The data are publicly available for further research upon request through the appropriate channels; raw data were deposited at the National Center for Biotechnology Information (NCBI) under the BioProject number PRJNA1261592.

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
