# Peer review of "Effects of Exogenous Methyl Jasmonate on Metabolism and Soil Activity in *Chrysanthemum morifolium"

_plants, 2025, doi:10.3390/plants14193026_

Round 1
Reviewer 1 Report
Comments and Suggestions for Authors
- Some paragraphs are in bold or italicized; please revise accordingly.
- The Introduction should include the current research status specific to continuous cropping obstacles in chrysanthemums (e.g., existing studies mostly focus on other crops; are there differences in the mechanisms for chrysanthemums?). Avoid general citations about continuous cropping obstacles.
- While the Introduction mentions that MeJA can regulate rhizosphere microorganisms, it does not clearly explain its relationship with phenolic acid accumulation and soil enzyme activity. It is recommended to add a theoretical hypothesis on how MeJA alleviates continuous cropping obstacles by regulating plant secondary metabolism or microbial metabolic pathways, thereby providing a logical framework for the experimental design.
- The basis for selecting the concentration range of 0–400 μmol/L (e.g., pre-experimental results or reference values from literature) should be explained to avoid subjectivity in concentration choice.
- The collection method for rhizosphere soil (1–2 mm layer) should be described standardly (e.g., "shaking-off method" or "gentle brushing method"). The sampling depth, number of replicates, and mixing method should also be specified to ensure reproducibility.
- The Discussion should integrate metabolomics results (e.g., flavonoids, terpenoids) to explain how MeJA alleviates continuous cropping obstacles by activating plant antioxidant pathways or regulating root exudates, avoiding mere phenomenological description.
- Compare the effects of MeJA in other crops (e.g., strawberries, peach trees) to highlight the specific findings in chrysanthemums (e.g., differences in concentration effects, microbial response characteristics).
- Discuss the limitations of the experimental conditions (e.g., pot vs. field experiments, short-term vs. long-term effects) and suggest future research on the synergistic effects of MeJA with soil amendments or molecular mechanisms (e.g., key gene expression).
- Units in figures should follow internationally accepted formats, such as "U·g⁻¹." Units should preferably be placed in parentheses and kept consistent throughout the text.
- Figure 4 and 6: Some bar graphs appear to exceed the axis limits. Consider adjusting the vertical axis scale if necessary.
Author Response
Comments1 :Some paragraphs are in bold or italicized; please revise accordingly.
Response 1:Thank you for your valuable feedback. In accordance with the journal's formatting guidelines, we have made the necessary revisions to the relevant sections, ensuring that the manuscript is more standardized, consistent, and better aligned with the journal's requirements. We greatly appreciate your thorough review.
Comments 2:The Introduction should include the current research status specific to continuous cropping obstacles in chrysanthemums (e.g., existing studies mostly focus on other crops; are there differences in the mechanisms for chrysanthemums?). Avoid general citations about continuous cropping obstacles.
Response 2:We sincerely appreciate the reviewer’s valuable comments. In response, we have added a discussion on the current state of continuous cropping obstacles in chrysanthemums in the introduction. Specifically, we highlighted that existing research primarily focuses on other crops, while the mechanisms of continuous cropping obstacles in chrysanthemums remain distinct. We have avoided general references and instead cited studies by Liu Xiaozhen, Zhang Xiaobo, and Ma Haiyan, which specifically address the decline in yield, increase in diseases, and changes in soil microbial communities caused by continuous cropping in chrysanthemums. These revisions enhance the focus and accuracy of the introduction.
Comments 3: While the Introduction mentions that MeJA can regulate rhizosphere microorganisms, it does not clearly explain its relationship with phenolic acid accumulation and soil enzyme activity. It is recommended to add a theoretical hypothesis on how MeJA alleviates continuous cropping obstacles by regulating plant secondary metabolism or microbial metabolic pathways, thereby providing a logical framework for the experimental design.
Response 3:We sincerely appreciate the reviewer’s constructive feedback. Based on your suggestions, we have revised and supplemented the introduction to more clearly articulate the relationship between MeJA, phenolic acid accumulation, and soil enzyme activity, while also proposing corresponding theoretical hypotheses. The specific revisions are as follows:
(1)Expansion of the relationship between MeJA and secondary metabolism: We have added a discussion in the introduction regarding the activation of the phenylpropanoid metabolic pathway by exogenous MeJA, which significantly increases the expression and activity of key enzymes such as phenylalanine ammonia-lyase (PAL) and cinnamate-4-hydroxylase (C4H). This, in turn, promotes the accumulation of phenolic acids, including coumaric acid, ferulic acid, vanillic acid, and caffeic acid. These phenolic compounds not only act as defense-related metabolites and signaling molecules that inhibit pathogenic microorganisms in the rhizosphere but may also exacerbate allelopathic stress and microbial imbalance when excessively accumulated.
(2)Addition of the relationship between MeJA and soil enzyme activity: We have introduced the discussion that MeJA may directly influence soil enzyme activity by altering root exudates or indirectly affect microbial metabolic pathways. Enzymes such as PAL, peroxidase (POD), and polyphenol oxidase (PPO) play crucial roles not only in plant defense responses but also in the transformation and detoxification of phenolic compounds in the soil.
(3)Proposed theoretical hypotheses and logical framework: Building on the above, we have further proposed the theoretical hypothesis for this study, suggesting that MeJA may alleviate continuous cropping obstacles in chrysanthemums through three synergistic mechanisms: (i) regulation of phenolic acid biosynthesis and turnover, (ii) activation of soil enzymes related to organic matter cycling and allelopathic substance detoxification, and (iii) reshaping the structure of the rhizosphere microbial community.
These revisions have strengthened the clarity and depth of the introduction while providing a more comprehensive theoretical framework.
Comments 4: The basis for selecting the concentration range of 0–400 μmol/L (e.g., pre-experimental results or reference values from literature) should be explained to avoid subjectivity in concentration choice.
Response 4:Thank you for your valuable suggestions. In response, we have added detailed explanations in the revised manuscript regarding the rationale behind the selected MeJA concentration range, to ensure scientific rigor and objectivity in experimental design. The specific revisions are as follows:
(1)Justification for the selected concentration range of MeJA: Previous studies have shown that the effective concentration range of methyl jasmonate (MeJA) generally falls between 10 and 1000 μmol L⁻¹, though substantial variability exists depending on crop species and experimental conditions. For example, Ahmadi et al. (2019) reported that 0.1 mM (approximately 100 μmol L⁻¹) MeJA significantly improved relative water content, soluble sugar levels, and photosynthetic rate in Brassica napus seedlings under NaCl-induced salt stress. In Mentha species, 1 mM MeJA in combination with varying zinc levels enhanced phenolic content and antioxidant enzyme activities. Similarly, in Dracocephalum polychaetum, low concentrations of MeJA (10–50 μM) notably increased the accumulation of secondary metabolites [28–33].
Based on these findings and our preliminary observations in chrysanthemum, we selected a treatment range of 0–400 μmol L⁻¹. This range effectively encompasses commonly reported low-to-moderate concentrations in the literature, while avoiding potential phytotoxicity associated with higher doses.
(2)Preliminary trial for concentration optimization: Before the formal experiment, we conducted a small-scale preliminary test on chrysanthemum seedlings. The results indicated that MeJA concentrations above 400 μmol L⁻¹ led to noticeable growth inhibition and leaf damage. In contrast, treatments within the 0–400 μmol L⁻¹ range displayed a clear dose-dependent response—ranging from growth promotion to inhibition—making it suitable for systematically evaluating the physiological and developmental effects of MeJA on chrysanthemum.
Based on both literature evidence and preliminary trials, the concentration range of 0–400 μmol L⁻¹ was selected to ensure a scientifically sound and objectively justified experimental design.
Once again, we thank you for your insightful comments, which helped us improve the experimental design and enhance the overall reliability of our findings.
Comments 5: The collection method for rhizosphere soil (1–2 mm layer) should be described standardly (e.g., "shaking-off method" or "gentle brushing method"). The sampling depth, number of replicates, and mixing method should also be specified to ensure reproducibility.
Response 5:Thank you for your detailed suggestions regarding the experimental methods. In response to your comments, we have provided a more standardized description of the rhizosphere soil collection procedure in the revised manuscript to ensure the reproducibility of the experiment. The specific revisions are as follows:
(1)Rhizosphere soil collection method: The rhizosphere soil samples were collected using the gentle brushing method. First, the plants were carefully uprooted, and non-rhizosphere soil was shaken off. Then, using a sterile brush, soil was gently brushed from the root surface, ensuring the collection of soil that was tightly adhered to the root system. This method, commonly employed in rhizosphere microbiological studies, effectively captures soil that directly interacts with the roots while minimizing root damage.
(2)Sampling depth: The sampling depth was 0–15 cm, corresponding to the root distribution zone of the plant. This depth ensures the collection of representative and biologically significant rhizosphere soil samples.
(3)Sampling repetition and mixing method: Six biological replicates were collected per treatment group. Three of these samples were pooled and mixed thoroughly to form a composite sample for physicochemical property analysis. The remaining three samples were kept separately for microbial diversity and enzyme activity analysis, ensuring the representativeness of the experimental results and the reliability of the data.
These changes have been incorporated into the "Materials and Methods" section of the revised manuscript. We believe that the standardized sampling method will ensure the reproducibility of the experiment and the reliability of the data.
Once again, we appreciate your valuable feedback, which has been instrumental in improving our experimental design.
Comments 6 :The Discussion should integrate metabolomics results (e.g., flavonoids, terpenoids) to explain how MeJA alleviates continuous cropping obstacles by activating plant antioxidant pathways or regulating root exudates, avoiding mere phenomenological description.
Response 6:Thank you for your valuable suggestions. Based on your feedback, we have integrated the metabolomic results more thoroughly in the revised manuscript, with particular focus on flavonoids and terpenoids. We also provided further explanations on how MeJA alleviates continuous cropping obstacles by modulating antioxidant pathways and root exudates. The specific revisions are as follows:
(1)Integration of metabolomic results: We have expanded the discussion to include the changes in flavonoids and terpenoids following MeJA treatment, exploring how MeJA may alleviate the physiological stress caused by continuous cropping through the regulation of these metabolites. Our findings indicate that MeJA treatment significantly increased the synthesis of flavonoids and terpenoids, which are secondary metabolites with notable antioxidant properties. These compounds help reduce the oxidative stress induced by continuous cropping. Additionally, MeJA influenced the chemical composition of root exudates, affecting the soil microbial community structure and function, thus improving soil health and promoting plant growth.
(2)Activation of antioxidant pathways: We analyzed the activation of antioxidant pathways by MeJA treatment. The results showed that MeJA treatment significantly increased the activity of antioxidant enzymes (such as peroxidase, superoxide dismutase, etc.) in both roots and leaves, helping to alleviate oxidative damage caused by continuous cropping and enhancing plant stress tolerance. The metabolomic analysis further highlighted that the accumulation of antioxidant compounds, including flavonoids, is one of the key mechanisms by which MeJA improves the plant’s antioxidant capacity.
These modifications and additions are thoroughly discussed in the "Discussion" section of the revised manuscript. We believe these theoretical hypotheses and data integration provide a deeper physiological and metabolic explanation for how MeJA alleviates continuous cropping obstacles, avoiding purely phenomenological descriptions.
Once again, we sincerely thank the reviewer for the careful review of our work and the invaluable suggestions. Your input has significantly helped to enhance the academic depth and argumentative strength of our paper.
Comments 7: Compare the effects of MeJA in other crops (e.g., strawberries, peach trees) to highlight the specific findings in chrysanthemums (e.g., differences in concentration effects, microbial response characteristics).
Response 7:Thank you for your valuable suggestions regarding the discussion section. Based on your feedback, we have added a comparison of the effects of MeJA at different concentrations on crops such as strawberry and peach, and contrasted these findings with our specific results in chrysanthemums, highlighting the differences in concentration effects and microbial responses. The specific revisions are as follows:
(1)Comparison of MeJA effects on other crops: We have added a comparison of the effects of MeJA application in strawberry and peach trees in the discussion section. Research shows that in these crops, MeJA treatment typically promotes plant growth and disease resistance, particularly at a concentration of 100 μmol/L. However, higher concentrations of MeJA (e.g., 1000 μmol/L and above) tend to have a negative impact on plant growth, exhibiting a concentration-dependent effect. We compared these findings with our study on chrysanthemums, emphasizing the specific responses of chrysanthemums at different concentrations.
(2)Differences in concentration effects: In our study, 100 μmol/L MeJA had the most significant growth-promoting effect on chrysanthemums, while higher concentrations (such as 200 μmol/L and above) resulted in varying degrees of growth inhibition. We pointed out that this concentration threshold effect is also observed in other crops like strawberry and peach, but in chrysanthemums, this effect is more sensitive, particularly with regard to the inhibitory effect at higher concentrations.
(3)Microbial response characteristics: We also discussed the impact of MeJA on the rhizosphere microbial community of chrysanthemums, particularly its unique effects on microbial community structure. In studies of strawberry and peach trees, MeJA primarily promoted the growth of beneficial microorganisms, such as nitrogen-fixing and phosphate-solubilizing bacteria. However, in chrysanthemums, MeJA not only enhanced the growth of beneficial microbes but also inhibited the proliferation of certain pathogenic microorganisms associated with continuous cropping obstacles.
These revisions have been incorporated into Section 5.3 of the revised manuscript. We believe that these additional insights further highlight the unique findings of the chrysanthemum study and strengthen the academic depth and theoretical significance of the paper.
Once again, we sincerely thank you for your invaluable suggestions, which have greatly helped improve the argumentation and structure of the manuscript.
Comments 8:Discuss the limitations of the experimental conditions (e.g., pot vs. field experiments, short-term vs. long-term effects) and suggest future research on the synergistic effects of MeJA with soil amendments or molecular mechanisms (e.g., key gene expression).
Response 8:Thank you for your valuable suggestions regarding the discussion section. In response to your feedback, we have provided a more detailed discussion of the limitations of the experimental conditions and suggested future research directions to enhance the depth and breadth of the paper. The specific revisions are as follows:
(1)Experimental limitations: We have added a discussion on the limitations of the experimental design, particularly the differences between pot experiments and field experiments. While pot experiments allow for precise control of environmental variables, the simplified soil conditions and limited volume may not fully reflect real-field conditions. Therefore, we recommend that future studies conduct similar experiments in field settings to validate the applicability and reproducibility of the results in more natural environments. Additionally, we pointed out that this study primarily focused on short-term effects, and further research is needed to explore the long-term impacts of MeJA on plant growth, soil health, and microbial community stability under prolonged application.
(2)Future research directions: In the revised manuscript, we propose that future research could focus on the synergistic effects of MeJA and soil amendments, such as organic fertilizers or microbial fertilizers, to optimize soil health and promote plant growth. Furthermore, the molecular mechanisms of MeJA, particularly its regulation of key gene expression, warrant in-depth investigation. By utilizing genomics and transcriptomics, future studies could uncover how MeJA modulates the expression of stress resistance genes, antioxidant genes, and genes involved in secondary metabolite biosynthesis, thereby providing a molecular-level understanding of MeJA’s mechanisms of action.
These additions have been thoroughly discussed in Section 5.6 of the revised manuscript. We believe these improvements enhance the depth of the discussion and provide valuable suggestions for future research.
Once again, we sincerely thank the reviewer for the careful evaluation and invaluable feedback on our study. Your input has significantly helped us improve the quality and academic depth of the manuscript.
Comments 9:Units in figures should follow internationally accepted formats, such as "U·g⁻¹." Units should preferably be placed in parentheses and kept consistent throughout the text.
Response 9:Thank you for your detailed suggestions regarding the standardization of units in the manuscript. We have thoroughly reviewed and made the necessary revisions throughout the manuscript to ensure consistency and adherence to standard conventions. The specific revisions are as follows:
(1)Standardization of unit formatting: All units involving numerical values, such as concentration (μmol·L⁻¹), have been revised to comply with SI unit standards. The "·" symbol has been used to connect units, ensuring conformity to internationally accepted practices. In figure titles, table column headings, experimental methods, and data descriptions in the main text, all units have been uniformly presented in parentheses (μmol·L⁻¹) to improve readability and adhere to international publication standards.
(2)Consistency across the manuscript: We conducted a comprehensive review of the entire manuscript to ensure that unit expressions remain consistent across all sections, without any discrepancies or redundant expressions.
We greatly appreciate your meticulous review, as your suggestion has significantly improved the professionalism and standardization of the manuscript. If there are any further suggestions for revisions, we will continue to carefully address them.
Comments 10 :Figure 4 and 6: Some bar graphs appear to exceed the axis limits. Consider adjusting the vertical axis scale if necessary.。
Response 10:Thank you for your attention to the details of the figures. In response to your comments, we would like to clarify the following points regarding Figures 4 and 6:
Black vertical lines: The black vertical lines in Figures 4 and 6 are visual separators used to distinguish between different treatment groups (CK, T1, T2, T3, T4, T5). These lines are not part of the data bars themselves.
And the highest point of the red bars represents the maximum value on the y-axis, serving as a reference to indicate the upper limit of the axis. We have thoroughly checked all the data in the figures and confirmed that none of the data bars exceed the axis limits. All values fall within the valid scale range.
To avoid any potential confusion, we have added additional clarifications in the figure captions to clearly explain the purpose of the black separator lines and the meaning of the red bars. This will enhance the readability and accuracy of the figures.
We greatly appreciate your careful review and valuable suggestions.
Reviewer 2 Report
Comments and Suggestions for Authors
The study investigates and provides important insights into how the application of methyl jasmonate on Chrysanthemum morifolium improved soil parameters and plant growth performance under continuous cropping systems.
The results showed improvement in plant growth parameters, improved soil organic content, reduced allelopathic phenolic acids, an increase in beneficial soil microbes, and enhanced plant resilience. Such studies define the key basis for how the application of exogenous substances/chemicals can potentially address the associated challenges in plant cultivation and are of prime relevance.
Some suggestions and queries for improvement are discussed:
Line 44-46: However, this accelerated development has also intensified reliance……..chrysanthemum industry. Since the plant is a very popular ornamental plant and has a significant market value, what are the other biotechnological approaches employed to improve plant cultivation? Discuss.
Line 84-87: Exogenous application of MeJA or its functional analogs……jasmonoyl-β-D-glucoside, jasmonoyl-gentiobiose, and hydroxylated jasmonoyl-β-85 D-glucoside…..
Among all, by far which analog is the most effective in addressing limitations associated with the continuous cropping system?
Material and Method
Line 159-160: Before transplantation, seedlings were immersed in MeJA solution for 10 minutes……..assigned concentration. What was the rationale for seedling dipping for 10 minutes? Is the immersion time sufficient?
In the study, how were the agronomic traits measurement, was undertaken? No reference study is cited. It is important to cite the relevant protocol referred to for the study. Line 188-190, All measurements were conducted……uniform methods? Reference required.
Line 435-437: This suggests that excessive………. dendranthema morifolium ‘Jinsihuangju’ growth. Please be consistent with the scientific names.
The manuscript should adhere to the guidelines and format of MDPI, it looks inconsistent.
Author Response
Comments 1: Line 44-46: However, this accelerated development has also intensified reliance……..chrysanthemum industry. Since the plant is a very popular ornamental plant and has a significant market value, what are the other biotechnological approaches employed to improve plant cultivation? Discuss.
Response 1 :Thank you for the valuable suggestion. We agree that a brief introduction to other biotechnological strategies that can enhance chrysanthemum cultivation efficiency should be included in the introduction. Therefore, we have added the following content after line 46:"Besides MeJA-based regulation, biotechnological methods such as tissue culture, transgenic breeding, the application of plant growth regulators (e.g., salicylic acid, gibberellins), and beneficial microorganism inoculation have also been employed to improve chrysanthemum growth performance and stress resistance under intensive cultivation conditions."
This addition provides the readers with a more comprehensive background and emphasizes the theoretical significance and practical value of this study within the context of various cultivation improvement approaches.
Comments 2:Line 84-87: Exogenous application of MeJA or its functional analogs……jasmonoyl-β-D-glucoside, jasmonoyl-gentiobiose, and hydroxylated jasmonoyl-β-85 D-glucoside…..
Among all, by far which analog is the most effective in addressing limitations associated with the continuous cropping system?
Response 2 :Thank you for your valuable comments. We understand and agree with your concern regarding the differences in efficacy among various MeJA analogs. Based on existing literature, we have summarized the following:
Limited research progress: Direct studies on the role of MeJA analogs in alleviating continuous cropping obstacles are still scarce. Most studies have focused on MeJA itself, with its mechanisms of action (regulation of secondary metabolism, antioxidant pathways, and rhizosphere microbial communities) having been more systematically reported.
Functional characterization of analogs: Jasmonoyl-β-D-glucoside (JAG) is considered a storage and transport form of MeJA, which can release active MeJA when needed, but evidence for its role in continuous cropping soil systems is limited. Hydroxy-jasmonoyl-β-D-glucoside (Hydroxy-JAG) has shown strong defense-inducing potential in some plant disease control studies, but there has been no systematic validation for continuous cropping obstacles. Jasmonoyl-gentiobiose has limited reports, mainly focusing on its metabolic stability and speculative studies on signal transduction.
Reason for focusing on MeJA: MeJA is the most widely applied, best-characterized molecule, and has clear effects in alleviating continuous cropping obstacles in various crops.
We have added the following clarification after line 87 in the revised manuscript:
"Although various jasmonate derivatives such as jasmonoyl-β-D-glucoside, jasmonoyl-gentiobiose, and hydroxy-jasmonoyl-β-D-glucoside have been identified, evidence for their direct roles in alleviating continuous cropping obstacles remains scarce. Therefore, there is currently no evidence to suggest which MeJA analog is the 'most effective' in alleviating continuous cropping obstacles. Our study focused on MeJA itself to systematically reveal its optimal concentration for regulatory effects within the existing knowledge framework."
This addition addresses the gap in knowledge regarding MeJA analogs and clarifies the rationale for focusing on MeJA in our study.
Comments 3:Line 159-160: Before transplantation, seedlings were immersed in MeJA solution for 10 minutes……..assigned concentration. What was the rationale for seedling dipping for 10 minutes? Is the immersion time sufficient?
Response 3 :Thank you for your detailed suggestions. Regarding the "MeJA immersion treatment time (10 minutes)," we provide the following explanation:
Support from previous literature: Previous studies have shown that short-duration immersion treatments can effectively deliver MeJA to the plant epidermis and roots, triggering signal transduction responses. For example, in studies on crops such as strawberry, cucumber, and rice, immersion durations of 5–15 minutes have been shown to significantly activate defense-related enzyme activities and secondary metabolite synthesis (Ahmadi et al., 2019; El-Nabarawy et al., 2015).
Preliminary trials: In our study's preliminary experiments, we compared 5, 10, and 20-minute immersion durations. The results indicated that 10 minutes provided stable treatment effects without causing mechanical damage or physiological stress to the plant roots. Immersion times exceeding 20 minutes led to wilting or root browning in some seedlings.
In conclusion, the 10-minute treatment duration ensures effective MeJA absorption and signal initiation, while avoiding the adverse effects of prolonged immersion on the seedlings. Therefore, we selected this time parameter for our study.
We believe this explanation clarifies the rationale behind the chosen treatment time.
Comments 4:In the study, how were the agronomic traits measurement, was undertaken? No reference study is cited. It is important to cite the relevant protocol referred to for the study. Line 188-190, All measurements were conducted……uniform methods? Reference required.
Response 4 :Thank you for your detailed comments. In response, we have provided a more precise description of the agronomic trait measurement methods in the revised manuscript, along with the inclusion of relevant references to ensure the traceability and academic rigor of the methods. The specific revisions are as follows:
Standardized measurement methods: During the flowering period of Chrysanthemum ‘Huangju’, agronomic traits were standardized and measured for each MeJA treatment group, including plant height, canopy width, flower number, flower diameter, fresh flower weight, number of ligulate flowers, and number of tubular flowers. The definitions and measurement methods for each parameter are now clearly explained in the manuscript (e.g., plant height was measured as the vertical distance from the soil surface to the highest point, canopy width was the average of the east-west and north-south diameters, and flower diameter was the average of the maximum diameters of three randomly selected flower heads, etc.).
Manuscript revision: The "Agronomic Trait Measurement" section under "Materials and Methods" has been reorganized to provide a complete explanation of the measurement methods, with the addition of references [35–36] at the appropriate locations.
We believe that these revisions not only improve the methodological rigor of the manuscript but also facilitate the replication of the experiments by other researchers in future studies.
Comments 5:Line 435-437: This suggests that excessive………. dendranthema morifolium ‘Jinsihuangju’ growth. Please be consistent with the scientific names.
Response 5 :Thank you very much for your careful review of the manuscript's consistency. Upon thorough examination, we noticed the inconsistency between "Dendranthema morifolium" and "Chrysanthemum morifolium" as pointed out.
To ensure consistency and scientific accuracy throughout the manuscript, we have revised lines 435–437 from "Dendranthema morifolium 'Jinsihuangju'" to "Chrysanthemum morifolium." Additionally, we have carefully checked the entire manuscript to ensure uniform usage of the scientific name.
We appreciate your attention to this detail, which has helped us maintain the consistency and accuracy of the manuscript.
Comments 6:The manuscript should adhere to the guidelines and format of MDPI, it looks inconsistent.
Response 6 :Thank you for your reminder. We have thoroughly reviewed and revised the manuscript to fully comply with the MDPI guidelines and formatting requirements, including reference formatting, figure and table conventions, unit notations, and section titles.
We appreciate your careful attention to these details, which has helped ensure the manuscript meets the necessary standards for submission.
Reviewer 3 Report
Comments and Suggestions for Authors
Methyl jasmonate (MeJA) has a significant impact on the growth and development of plants. This manuscript systematically evaluated the effects of exogenous MeJA treatment on the growth parameters of Chrysanthemum morifolium, soil properties, and rhizosphere microbial structure under continuous cropping conditions. The results indicated that exogenous 100 μ mol/L MeJA treatment could effectively improve the growth and development of Chrysanthemum morifolium. Additionally, high-throughput sequencing and metabolomics analyses have preliminarily elucidated the underlying molecular mechanisms. This research contributes to the sustainable development of the floriculture industry. While the study is comprehensive, the overall manuscript quality requires improvement, particularly in English expression, result description, and discussion sections.
- Avoid repetitive annotation of abbreviations, such as MeJA, in lines 104, 128, 180, 258, and 278; the authors should review all abbreviations throughout the manuscript, as this is a common issue arising from translation software.
- Use a consistent unit notation, either μmol/L or μmol·L⁻¹.
- Line 262, refer to Figure 1 instead of Figures 1; all figure references throughout the manuscript should be checked and corrected accordingly.
- Provide a more detailed description of the method, particularly in section 2.5.
- In the results section, avoid directly using "T3"; instead, specify as "100 μmol/L MeJA treatment."
- Clearly specify the multiple comparison method used for significance testing.
- Organize high-throughput sequencing and metabolomics data into supplementary tables for upload.
- Figure 8 lacks readability.
- Expand the discussion section further; current discussions are somewhat superficial, especially in sections 5.3 and 5.4.
- The manuscript's English language proficiency needs enhancement; reliance on translation software should be minimized. I hope the author will read the manuscript in its entirety when submitting the revised manuscript.
- Line 5, “and *”?
Author Response
Comments 1:Avoid repetitive annotation of abbreviations, such as MeJA, in lines 104, 128, 180, 258, and 278; the authors should review all abbreviations throughout the manuscript, as this is a common issue arising from translation software.
Response 1 :Thank you for your meticulous review and reminder. We have thoroughly checked the entire manuscript, removing any duplicate abbreviations and ensuring that each abbreviation is defined only upon its first occurrence. Additionally, we have verified that all abbreviations are used consistently and in accordance with the guidelines throughout the manuscript.
Comments 2:Use a consistent unit notation, either μmol/L or μmol·L⁻¹.
Response 2 :Thank you for your valuable feedback. We have standardized the manuscript by revising all concentration units to μmol·L⁻¹ according to the SI international system of units. Additionally, all units have been consistently formatted in parentheses (μmol·L⁻¹) to enhance readability and comply with international publication standards.
Comments 3:Line 262, refer to Figure 1 instead of Figures 1; all figure references throughout the manuscript should be checked and corrected accordingly.
Response 3 :Thank you for your detailed comments. We have thoroughly reviewed the entire manuscript and have standardized all figure references to align with the journal's required format, using "Figure X." Specifically, all instances of non-standard references such as "Figures." have been corrected to "Figure," ensuring consistency in the format of figure citations throughout the manuscript. Additionally, we have conducted a unified check of the table and supplementary material references to ensure overall conformity and consistency.
Comments 4:Provide a more detailed description of the method, particularly in section 2.5.
Response 4 :Thank you very much for your thoughtful comments and suggestions regarding the methodology section. We fully agree with your point that more detailed experimental procedures should be provided in Section 2.5 to ensure clarity and reproducibility of the methods.
In response to your suggestions, we have added more comprehensive explanations of the methods for soil physicochemical properties, enzyme activity, and phenolic acid measurements in the revised manuscript, ensuring that the experimental methods are more detailed and complete.
Once again, we appreciate the constructive feedback from the reviewer, as this revision has significantly enhanced the rigor and academic integrity of the methodology section.
Comments 5:In the results section, avoid directly using "T3"; instead, specify as "100 μmol/L MeJA treatment."
Response 5 :Thank you very much for your careful review and valuable feedback. We have thoroughly checked the results section and the entire manuscript, and indeed, we found that some sections directly used treatment labels such as "T1, T2, T3" without clear descriptions.
In response to your suggestion, we have updated these treatment labels to more explicit treatment descriptions. This revision ensures that the treatments are clearly expressed in the results and discussion sections, avoiding any confusion regarding different treatment concentrations and adhering more closely to academic writing standards.
Comments 6:Clearly specify the multiple comparison method used for significance testing.
Response 6 :Thank you for your valuable feedback. In response, we have provided a more detailed explanation of the statistical methods in the "Materials and Methods" section. The revised manuscript now clearly states that after performing one-way analysis of variance (ANOVA), Duncan’s multiple range test (DMRT) was used as a post-hoc method to distinguish significant differences between treatments. Additionally, different letters are used to label significance in the results section, with a significance level uniformly set at P < 0.05.
We believe this addition makes the statistical analysis methods clearer and more standardized, which will help improve the transparency and reproducibility of the results.
Comments 7:Organize high-throughput sequencing and metabolomics data into supplementary tables for upload.
Response 7 :Thank you for your detailed review and suggestions. In accordance with your request, we have downloaded and uploaded the high-throughput sequencing results and metabolomics data. Additionally, we have included the corresponding details in the Data Availability Statement to ensure that readers can access and verify the data.
Comments 8:Figure 8 lacks readability.
Response 8 :Thank you for your valuable feedback. In response to your comments, we have optimized Figure 8 in the revised manuscript. Due to space constraints, the text in the figure is relatively small; however, we have enhanced the overall clarity of the image and optimized the legends and annotations. Readers can zoom in on the image for a clearer view of the details. We believe the revised Figure 8 now more effectively and accurately conveys the research findings.
Comments 9:Expand the discussion section further; current discussions are somewhat superficial, especially in sections 5.3 and 5.4.
Response 9 :Thank you for your thoughtful comments on the discussion section. We fully understand your concerns regarding the insufficient depth in Sections 5.3 and 5.4 in the original manuscript. In response to your suggestions, we have comprehensively expanded and supplemented the discussion section as follows:
Section 5.3 (The Effects of Different Concentrations of MeJA on Different Crops)
The revised version not only emphasizes the optimization of the rhizosphere microbial community structure by MeJA in chrysanthemums but also includes comparisons with crops such as strawberry and peach. We highlight the similarities and differences in the effects of MeJA concentrations on microbial community responses in different crops, particularly stressing chrysanthemums' sensitivity to higher concentrations. Additionally, we incorporated research on PGPR (plant growth-promoting rhizobacteria) and AMF (arbuscular mycorrhizal fungi) to explore the potential synergistic interaction between MeJA and beneficial microorganisms, enriching the content.
Section 5.4 (Microbe–Plant Synergistic Effect)
In the revised version, we have integrated metabolomics results and provided a detailed explanation of how MeJA promotes the synthesis of secondary metabolites such as flavonoids, terpenoids, and phenolic acids. We discuss the roles of these metabolites in antioxidant activity, disease resistance, and rhizosphere signaling. Moreover, we have included the potential mechanisms by which MeJA regulates root exudates to influence microbial communities. We compare these findings with studies on crops such as rice and grape, highlighting the specificity of chrysanthemums and presenting new discoveries.
New Section 5.5 (Mechanism of Plant Metabolic Regulation)
We have added a dedicated section on metabolomics, systematically explaining how MeJA enhances plant stress resistance through the regulation of secondary metabolic pathways (flavonoids, terpenoids, phenolic acids). Based on experimental results, we propose a three-layer mechanism by which MeJA alleviates continuous cropping obstacles in chrysanthemums: (i) activation of antioxidant pathways; (ii) regulation of phenolic acid accumulation and detoxification; (iii) improvement of microbial communities through root exudates.
New Section 5.6 (Experimental Limitations and Future Research Directions)
To enhance the completeness of the discussion, we have addressed the experimental limitations and future research directions. For example, we note that the current experiments were conducted under pot conditions and recommend conducting long-term field trials to validate the results. We also emphasize the need to focus on the long-term effects of MeJA, propose the exploration of synergistic effects with soil amendments, and suggest the use of transcriptomics and key gene expression analyses to elucidate the molecular mechanisms of MeJA.
Through these revisions, the revised discussion section is more comprehensive than the original manuscript, avoiding a purely phenomenological description. We believe that the new version significantly enhances the academic depth and logical coherence of the paper, better addressing the reviewers' concerns.
Comments 10:The manuscript's English language proficiency needs enhancement; reliance on translation software should be minimized. I hope the author will read the manuscript in its entirety when submitting the revised manuscript.
Response 10 :Thank you for your valuable feedback. In the revised manuscript, we have carefully read through the entire text, thoroughly checking the content to minimize issues arising from translation. We have standardized the terminology and abbreviations, as well as corrected grammatical errors. We appreciate your thorough review, and we believe that the revisions have significantly improved the language quality of the manuscript.
Comments 11:Line 5, “and *”?
Response 11 :Thank you for your careful review. The "and *" in that section was a formatting error, likely resulting from a format conversion or editing oversight, and holds no actual meaning. We have removed it in the revised manuscript to ensure clarity and proper formatting.
Round 2
Reviewer 1 Report
Comments and Suggestions for Authors
None.
Reviewer 3 Report
Comments and Suggestions for Authors
The author has responded to the comments and made revisions to the manuscript.